# Investigation of an Innovative Roll-to-Plate (R2P) Hot-Embossing Process for Microstructure Arrays of Infrared Glass

**DOI:** 10.3390/mi15111307

**Published:** 2024-10-28

**Authors:** Qinjun Li, Kangsen Li, Jinyu Lv, Linglong Tao, Feng Gong

**Affiliations:** 1Shenzhen Key Laboratory of High-Performance Nontraditional Manufacturing, College of Mechatronics and Control Engineering, Shenzhen University, Shenzhen 518060, China; liqinjun2022@email.szu.edu.cn (Q.L.); likangsenszu@163.com (K.L.); 2410095128@mails.szu.edu.cn (J.L.); 2410095119@mails.szu.edu.cn (L.T.); 2Guangdong Provincial Key Laboratory of Micro/Nano Optomechatronics Engineering, College of Mechatronics and Control Engineering, Shenzhen University, Shenzhen 518060, China

**Keywords:** morphological regulation, roll-to-plate hot embossing, microstructure arrays, micro-manufacturing, micro-featured surface on infrared glass

## Abstract

The roller-to-plate (R2P) hot-embossing process is an effective, low-cost method for producing high-quality micro-/nano-optical components. In the field of night vision applications, the fabrication of chalcogenide glass microstructures is emerging as a promising alternative to traditional infrared glass. This trend is driven by the potential of chalcogenide glass to surpass conventional materials in terms of performance. However, the development of R2P hot embossing faces challenges, such as the high cost of curved mold manufacturing, the reliance on roll-to-roll processes for nano hot embossing, the limitations of plastic materials, and the unclear viscoelastic properties of infrared glass. In this study, a novel R2P hot-embossing process was developed to fabricate flat chalcogenide glass structures. The key parameters, such as roller temperature, speed, and embossing pressure, were investigated to understand their impact on the glass-filling performance. The deformation mechanism of the glass microstructures was also analyzed. The experimental results show that the R2P hot-embossing method offers excellent reproducibility, achieving a maximum filling rate of 96% and an average roughness deviation of 8.36 nm. The increase in the roller temperature and embossing force increased the filling height of the glass microstructure arrays, while an increase in the roller speed decreased the filling height. Different embossing methods, including variations in speed, temperature, and force, are summarized to analyze the structural changes during embossing. This study provides a foundation and a basis for future research on the roller-to-plate hot embossing process.

## 1. Introduction

Infrared and night vision technologies have been widely used in national defense and industry and in various professions [1,2,3]. Infrared optical lenses are crucial for systems such as infrared temperature measurement, thermal imaging, medical monitoring, and heart rate detection [4]. Devices with microstructures, such as microfluidic chips [5], microlens arrays [6], Fresnel lenses [7], and gratings [8], are being used more and more widely; the demand for microstructure-enhanced infrared glass in the field of infrared imaging and the range of its applications are increasing. The traditional manufacturing of microcylindrical infrared glass lenses involves complex processes, such as milling, fine grinding, polishing, and coating. These steps extend the processing time, and they also increase the risk of installation errors, thereby reducing efficiency and molding accuracy. As a result, there is a growing need for more streamlined and precise fabrication methods to meet the rising demand for microstructure-enhanced infrared glass.

To address these challenges, researchers have developed advanced machining processes aimed at enhancing manufacturing efficiency and accuracy while reducing polishing time. Techniques such as ultra-precision cold machining technology, single-point diamond turning [9], milling [10,11], grinding [12,13], and micro-fly cutting [14,15] have been explored. Although these methods achieve high precision in surface shaping, they often fall short in meeting market demands for cost-effectiveness and scalability.

R2P hot embossing is commonly used in the hot embossing of optical polymers and is an expansion of hot embossing technology. Infrared lenses with microstructures have significant advantages, such as their light weight, large aperture, low thickness, low cost, and good light-gathering effects. For this reason, the research on the key technology aspects of the ultra-precision hot embossing of red microstructure lenses has very important academic value and broad engineering application prospects. The R2P process was initially developed in 1998, and Tan and Chou [16] proposed roller nanoimprint lithography (RNIL), which is a roller nanoimprinting system that has allowed pattern transfer with a resolution of less than 100 nm. Chang et al. [17] investigated the microstructure using a filling process in the roll-to-roll process. Thin steel molds with micro-circular whole-array patterns were prepared using photolithography and wet chemical etching processes. The thin steel molds were then wrapped around metal cylinders to form embossed roll molds. Under appropriate processing conditions, the molten polymer will only partially fill the micro-circular holes of the mold and form a convex lens surface due to surface tension. Continuous plastic optical films with microlens array patterns were obtained. Lee et al. [18] derived an analytical model for polymer flow cavity filling during R2P embossing based on the Hertzian contact pressure distribution and the Navier–Stokes equation. The feasibility of this model was assessed using a lab-scale prototype of the R2P microthermal embossing system for the large-area replication of micropatterns on polymer substrates. However, there is a lack of research related to roll embossing for infrared glass due to the challenges involved in embossing glass with equipment that has a structure made of oxidation-resistant, low-vibration, and high-precision materials. The deformation theory and modeling of anti-reflection glass is still incomplete, especially regarding the filling deformation mechanism at the micrometer scale; these are all aspects that need to be further extended and studied in depth.

The optical profile deviation of hot-press-formed lenses is one of the key factors affecting the optical path deviation and its imaging quality. Su et al. [19] used a mold-compensation correction strategy to reduce the profile deviation of hot-press-formed lenses and measured the shape profile dimensions of the lenses using a Talysurf profilometer; the experimental results show that the geometrical profile deviation of the lenses was less than 1.5 μm. Ananthasayanam et al. [20,21] analyzed the evolution of the shape deviation of different types of lenses based on the viscoelastic properties of glass materials and discussed the influence of the hot press process parameters on the shape deviation of the formed lenses, and they summarized the sensitivity coefficients of the parameters. Zhang et al. [9] machined a microlens array unit with a diameter of 601 μm and a depth of 5.22 μm on the surface of an aluminum mold using a slow-tool servo diamond turning technique; they also formed a surface roughness of 5.22 μm using hot pressing to produce a microlens array unit. Additionally, infrared microlens arrays with a surface roughness of 15.61 nm and a topographic deviation of 60 nm were formed using hot-press molding.

Kim et al. [22] developed a method to transfer the Fresnel shape from a nickel mold, processed using single-point diamond turning, to a glassy carbon mold through a series of three inverted mold replication processes. This approach resulted in approximately 22.5% shrinkage of the Fresnel glassy carbon mold structure during the heating and carbonization stages. However, a compensation algorithm was implemented to reduce the shape deviation of the glassy carbon mold structure obtained through inverted molding to 3 μm, while the shape of the resultant glassy Fresnel lens achieved a deviation of 4 μm after hot pressing. The optical Fresnel elements produced by this method are well suited to non-imaging optical systems. To ensure high optical imaging quality, the molding accuracy of the optical elements is generally required to be within one-quarter of the applied wavelength. Therefore, precise control over the shape deviation of the glass lens during the hot-embossing process is crucial. While the research highlights the significance of replication accuracy in the optical elements, a comprehensive analysis indicates that the existing methods do not adequately address the force deformation phenomena of the mold material or the mechanisms of glass-filling deformation during the forming process. Consequently, despite the advancements in hot-embossing techniques, unresolved challenges remain in the production of high-precision glass optical elements.

In this study, we utilized a self-developed roller-to-plate (R2P) hot embossing machine to conduct controlled variable experiments aimed at investigating the effects of three key parameters: embossing temperature, embossing speed, and embossing force. Multiple sets of repeatability verification experiments were performed to ensure the reliability of the results. The 3D morphology and surface quality of the embossed glass microstructures were characterized using scanning electron microscopy (SEM) and white light interferometry. Ultimately, this study reveals the underlying mechanisms of the R2P hot-embossing process for glass, providing a theoretical foundation for future research and development in this area. The insights gained from this work are expected to contribute to the optimization of glass microstructure fabrication and to enhance the overall efficiency and quality of the embossing process.

## 2. Materials and Methods

### 2.1. Glass and Mold

In order to meet the demand for high-quality imaging for infrared night vision, accurate temperature detection, and photoreceptor robustness, the mold materials used for roll-to-plate hot embossing must have high hardness and strength, low coefficients of thermal expansion, and excellent high-temperature oxidation resistance. At the same time, it is also necessary to consider the unique thermal shock corrosion, high-temperature oxidation, high stress, and other harsh processing environments of the glass hot-embossing process.

The commonly used mold materials for the roll-to-plate hot-embossing process include advanced ceramics (such as boron nitride, silicon nitride, and silicon carbide), cemented carbide (including tungsten carbide and chrome–nickel alloys), and other materials (such as monocrystalline silicon and nickel phosphide). Among these, alumina stands out as one of the most promising options due to its exceptional properties. Its high hardness, excellent wear resistance, and outstanding oxidation resistance make it less susceptible to deformation and fracture during the hot-embossing process.

At high temperatures, sufficient mechanical strength and chemical inertness allow the glass to be realized to the greatest extent during the filling process. In this work, picosecond laser technology was applied to the fabrication of the molds for aluminum oxide. Alumina molds with apertures of 10 μm in diameter, 500 nm in depth, 10 mm in diameter, and 500 μm in thickness were used in the experiments. These were made using picosecond laser drilling in a mold. The polished glass used for the experiments in the hot-embossing process was an infrared glass with As_40_Se_60_ as the main component and the grade HSW6; it was supplied by Nantong Hua Heng Optical Glass Co. (Nantong, China). It had a transition temperature of 185 °C. The glass sample had a diameter of 9.7 mm and a thickness of 3 mm. Figure 1a shows the roll-to-plate hot-embossing process. In this setup, the displacement of the Z-axis allows the roller to approach the glass substrate. Both the roller and the glass move in the same direction and at the same linear speed, facilitating the embossing process. As the roller makes contact with the glass, a linear force is applied through the roller to the thimble, which effectively transfers the embossing force to the glass. This force initiates the embossing process, allowing the glass to be deformed incrementally. The glass undergoes partial embossing in a step-by-step manner until the desired embossed pattern is fully achieved.

Figure 1b,c shows the detailed surfaces of the alumina molds and embossed glass microstructures characterized using scanning electron microscopy (SEM) and white light interferometry. Picosecond lasers were used to machine the micro-via arrays on the surface of the sapphire molds, and Delong laser processing equipment DP100 was used. In the R2P hot-embossing process, temperature plays an important role in the deformation of the glass. Excessively high embossing temperatures tend to cause interfacial adhesion between the mold and the glass, while excessively low temperatures may increase the resistance of the glass to deformation and lead to difficulties in filling the microstructure. During the hot-embossing process, the optical glass and mold assembly are heated by an infrared lamp over a period of 300 s to 230 °C. After the temperature sensor fixed to the mold surface reaches the desired temperature, a soaking process lasting 180 s is performed to homogenize the glass temperature. Then, with the loading of the embossing roller, the embossing force is linearly applied to the surface of the glass, which is gradually and continuously filled in the direction of the hot embossing by this driving force; the gases are vented from the other side. At the end of the hot embossing stage, the annealing stage begins. Samples are taken in the next cooling stage after cooling with a furnace.

### 2.2. Experiment Design

The specific parameters of each group of experiments designed in the experiment are shown in the following Table 1.

### 2.3. Characterization

Figure 2 shows the morphology of the embossed glass, measured using a 3D profiler (S-nexo, Bruker, Madrid, Spain). Two scanning electron microscopes were used (Scios, FEI, Thermo fisher, Waltham, MA, USA; and Gemini SEM 560, Zeiss, Oberkochen City, Germany). In order to evaluate the processing accuracy of the roll-to-plate hot-embossing process, five zones were sampled to obtain data for each process parameter. The microstructure of each region was collected in 4 × 4 arrays to evaluate the microstructural quality. The evaluation consisted of the filling height complex regime and the average roughness (Sa), using two different sampling methods: the sampling of the roughness data at the lens and the overall sampling of the roughness data. The overall sample roughness (sampling area: 63 μm × 63 μm), the roughness of individual lenses (sampling area: 5 μm × 5 μm), and the average deviation of the height of the microlens and its average mold deviation were collected, respectively. In order to obtain a more accurate reproduction rate, we took three measurements at five positions for each bare glass sample and micrometer structure sample to calculate the average data deviation.

## 3. Results and Discussion

### 3.1. Replication Quality of Glass Microstructure Arrays

Five repetitive experiments were executed before the formal experiments. The embossing conditions were as follows: 240 °C embossing temperature, 150 N embossing force, and 0.2 mm/s embossing speed. After the end of embossing, the characterization method above was used. Figure 3a shows the overall roughness in the repetitive experiments, and Figure 3b shows the average deviation of the roughness of the individual microlenses and the average deviation of the roughness of the individual mold holes. The average deviation of the microstructure of the infrared glass surface from the surface roughness of the mold was 0.878 nm. The surface roughness of the embossed glass was well below the 10 nm deviation from the surface roughness required for general-purpose optical components. Figure 3c demonstrates the error of the average height deviation of the microstructure versus the average deviation of the mold depth for five sets of repeated experiments. Figure 3c shows an average deviation of 498.53 nm in the structure height with a replication rate of >95%. The results show that the height deviation of the mold and glass microstructures had a minimum of 3.16 nm, a maximum of 12.08 nm, and an average of 5.862 nm. The R2PHE process demonstrated satisfactory reproducibility.

Figure 3d,e shows the variation in the roughness of the overall and single lens microstructures at different embossing speeds. Together, they lead to an increase in the surface roughness of the glass. By observing the changes in the average roughness of the individual microlenses due to all the embossing speeds, it can be assumed that the average roughness of the individual microlenses increases with the increase in the embossing temperature. With the increasing embossing temperature, the IR glass and mold surface are under conditions of coupled thermal and force fields. The R2P hot-embossing process is conducted. On one hand, the glassy selenium arsenide gradually fills the molding. On the other hand, the molecular interpenetration becomes more intense between the selenium arsenide and aluminum oxide. Materials with different melting points undergo violent interpenetration. Consequently, the processing temperatures suitable for selenium arsenide may not be appropriate for aluminum oxide, leading to severe adhesion issues and potential defects during demolding. Additionally, inconsistent coefficients of thermal expansion can result in dimensional changes during rapid temperature fluctuations.

The average deviation of the roughness of the microlenses with respect to temperature was smallest for the experimental group at an embossing speed of 0.2 mm/s. The range deviation at this speed was the second lowest among all the data points. A lower maximum deviation indicates better uniformity of the microlenses. Therefore, an embossing speed of 0.2 mm/s was identified as having the highest potential to produce the best quality microlenses.

For all types of surface roughness, an increase in embossing temperature correlated with an increase in surface roughness across all the experimental groups (see Figure 3d,e). However, there appeared to be no consistent pattern in the fold lines for the different embossing speeds, which we attribute to the micro-vibrations of the drum and thimble during the embossing process. Additionally, a speed gradient that is set too low may also contribute to this inconsistency. When examining the overall roughness, separate analyses were conducted for speeds of 0.2 mm/s and 1.0 mm/s, as well as for 0.6 mm/s and 2.0 mm/s. The results indicate that as the speed decreased, the overall surface roughness increased, which is consistent with the structural height observed. Regarding the roughness of the individual microstructures, the surface roughness of the microlenses decreased with lower speeds. However, the experimental group at 250 °C did not conform to this trend; this finding is due to the adhesion phenomena, as explained below. In summary, a common observation in the experiments across all the types of roughness at different velocities is that lower temperatures result in lower roughness for individual microstructures.

Figure 3f shows the variation in the overall roughness of the microstructure and the average roughness of the individual microlenses under different embossing forces (other conditions were a 240 °C embossing temperature and a 0.2 mm/s embossing speed). The results indicate that both the overall microstructure roughness and the average roughness of the individual microlenses exhibited an increasing trend as the embossing force increased, suggesting a positive correlation between roughness and embossing force. The rise in embossing force enhanced the contact stress between the upper mold and the glass. At a constant embossing speed, which can be considered an equivalent of a low-frequency vibration frequency, the low-frequency stress amplitude per unit area of the microstructure increased. This resulted in a corresponding increase in both the overall and the local microstructure roughness. While this phenomenon was more pronounced in the overall roughness, its effect on the average roughness of individual microlenses was less obvious. It is important to note that the values presented in Figure 3f are ten times those of the original data; these numbers were calculated to facilitate a comparison between the overall and local roughness in a single figure.

Figure 4a shows three types of microstructures: a bamboo-shoot-like array of microstructures of less than 100 nm in height, a micro-lenticular array with a structural height between 100 nm and 300 nm, and fully filled microstructural arrays with a structural height greater than 300 nm. Figure 4b shows that the height of the microstructure increased from 307.67 nm to 351.04 nm as the embossing force increased from 50 N, with a pressure gradient of 50 N to 250 N. Figure 4c shows a comparison of the experimental structures for embossing speeds of 0.2 mm/s and 2.0 mm/s. The embossing speed affects the embossing time and the embossing force duration during the embossing process; therefore, the embossing speed and embossing temperature are theoretically coupled. The slower the embossing speed, the greater the contact time between the mold and the glass under the embossing force, and the greater the heat energy transferred through heat conduction, the more it can overcome the defects of the glass caused by the thermal creep effect of embossing. Regarding the microstructure height (at 210 °C and 220 °C embossing temperatures), the faster sample was taller. For the other samples (240 °C and 250 °C), adhesion was present (see Figure 5b,c). Figure 4d shows that the embossing speed increased from 0.2 mm/s to 1.0 mm/s, and the height of the embossing glass showed a trend indicating that the lower the embossing speed, the higher the structural height. The filling rate of the experimental group (0.2 mm/s, 240 °C) was greater than 90%, and the filling rate of the experimental group (0.2 mm/s, 250 °C) was lower than 80%, considering the phenomenon of adhesion and fracture. At the same time, at 250 °C, the results of all the experiments were close to 300 nm, and it can be assumed that the fracture occurring after fully filling the embossed flowers was similar.

Figure 5a shows the atomic energy spectrum analysis of the sapphire mold, while Figure 5b,c shows the atomic energy spectrum analysis of the sulfur glass prepared with this mold. The main component of the sulfur glass HSW6 is selenium arsenide (As60Se40), and its elemental composition appears to be reasonable. During the roller-to-plate hot-embossing process (conducted at an embossing temperature of 220 °C, a glass transition temperature of 185 °C, and a glass-softening temperature of 240 °C), high temperatures and applied embossing forces lead to a degree of interpenetration between the alumina and selenium arsenide. The presence of arsenic (As) and selenium (Se) elements in the mold, measured at 1% and 4%, respectively, suggests minimal adhesion between the alumina mold and the HSW6 sulfur glass. This indicates that even at elevated temperatures during the hot pressing of glass, the mold maintained good non-stick properties. In contrast, under another condition (embossing temperatures 250 °C), the analysis of the glass samples reveals Al and O contents of 10% and 2%, respectively. This suggests that the adhesion of the sulfur-based glass to the alumina mold was more significant, indicating that a greater amount of alumina material adhered to the surface of the glass under high pressure and temperature conditions.

### 3.2. Filling Mechanism and Morphology Evolution of Glass Microstructure Arrays

The morphology of the structures can be classified into three categories: flat-peak filling mode, single-peak filling mode, and residual-peak filling mode, based on the height and curvature of the edge profile. Figure 6a–c shows the 3D morphology and main features of the flat-peak filling mode. This mode typically arises under specific conditions, including low embossing temperatures (below 220 °C), high embossing speeds (exceeding 1.0 mm/s), or low embossing forces (below 100 N).

The embossing temperature is the most important parameter of R2P hot embossing. An appropriate temperature facilitates the deformation of glass, allowing effective embossing with minimal force and time. However, temperatures that are too high or too low can lead to complications. Excessive temperatures can cause significant interpenetration between the glass and the mold, resulting in adhesion issues. Conversely, low embossing temperatures may prevent the glass from exceeding its glass transition temperature, leading to high viscosity conditions. Under these circumstances, low embossing forces and rapid embossing speeds can result in limited material flow during the embossing process.

The embossing speed affects creep deformation during the glass embossing process. The creep effect, which is characteristic of glassy materials, indicates that achieving high-quality embossed reproductions requires sufficient holding time. During the hot embossing and filling of glass, the creep effect plays a crucial role; shorter embossing and holding times limit the height of the molded glass structures. Even if the embossing temperature or force is increased, a reduced embossing time constrains the process due to the creep effect, resulting in only low-height structural arrays being successfully embossed.

The force required for embossing microstructures is derived from both the self-weight of the glass and the applied embossing force. Glass flow can occur under its own weight when the embossing temperature exceeds the glass transition temperature. However, at this temperature, the viscosity of the glass complicates the dimensional control, making this approach generally unsuitable for precision hot-embossing processes. Consequently, the embossing force becomes a critical factor in determining the characteristics of the embossed microstructure. It can be concluded that the embossing force is positively correlated with the height of the microstructure. Additionally, it was observed that the low embossing force may have contributed to the occurrence of the flat-peak filling pattern.

The flat-peak filling pattern represents a low-height filling state in microstructural morphology; it is characterized by the minimal height of the microstructure and an almost vertical profile with no noticeable inclination. In contrast, the single-peak filling pattern allows the formation of a microcylindrical array structure with a defined slope. Figure 6d–f shows a summary of the most common surface morphologies and their characteristics for single-peak filling. The samples with single-peak filling patterns are most likely to appear under the conditions of a suitable embossing temperature (220 °C–230 °C), suitable embossing speed (0.2 mm/s–0.6 mm/s), and suitable embossing force (100 N–200 N). In the case of line contact, a certain vibrational frequency stress, a stable embossing force, and a suitable embossing temperature, the microstructures continuously undergo unimodal filling as the embossing progresses. Notably, significant shrinkage occurred around the circumference of the microstructure arrays when the single peaks were filled.

Figure 6g–i shows the residual-peak filling pattern; it is a pattern that summarizes a variety of extreme embossing states. This pattern can arise from several factors, including excessively fast embossing speeds that result in low structural depths. Additionally, high embossing temperatures can cause significant interpenetration between the mold and the glass, complicating the demolding process. Furthermore, excessive embossing forces may lead to permanent glass residue remaining in the mold cavities.

## 4. Conclusions

In this study, we developed a novel roll-to-plate hot-embossing process and conducted a series of non-contact hot embossing experiments for the fabrication of microlens arrays. The effects of the process parameters on microlens replication accuracy were investigated, and the uniformity and molding patterns of the microlenses were evaluated and summarized. The main results are as follows.

(1)Flat mold roll-to-plate thermal embossing is an effective method for preparing micro-/nanostructures. A variable embossing force makes the embossing process easier and shorter, with higher processing flux per unit.(2)The effects of the R2P hot embossing process parameters on the average deviation of the structure height, the overall roughness, and the average deviation of the individual microlens height of the microstructure arrays were investigated. In the replicate experimental group of the optimal process, the average filling rate of the structures was 96%, the maximum deviation of the mold from the glass was 12.08 nm, the average surface roughness of the glass was 201.52 nm, and the average deviation of the height of the individual microlenses was 8.36 nm.(3)Temperature was found to be the main factor affecting the results of the R2P heat embossing. The temperature was positively proportional to the microstructure height in the 210 °C–250 °C interval (150 N embossing force, 0.2 mm/s–1.0 mm/s embossing speed). The temperature was positively correlated with microstructure roughness (150 N embossing force, 0.2 mm/s embossing speed). Increasing the embossing speed improved the embossing efficiency, and embossing speeds less than 1 mm/s were considered reasonable (with a stable embossing height and embossing roughness RMS). The embossing force was positively and proportionally correlated with the structure height and roughness in the interval of 50 N–250 N (processing conditions: 0.2 mm/s embossing speed, 230 °C embossing temperature).(4)The filling modes of the microstructures were closely related to the temperature process parameters; second are pressure and speed. The filling behaviors during R2P thermoforming were mainly in the flat-peak filling mode, single-peak filling mode, and residual-peak filling mode.

## Figures and Tables

**Figure 1 micromachines-15-01307-f001:**
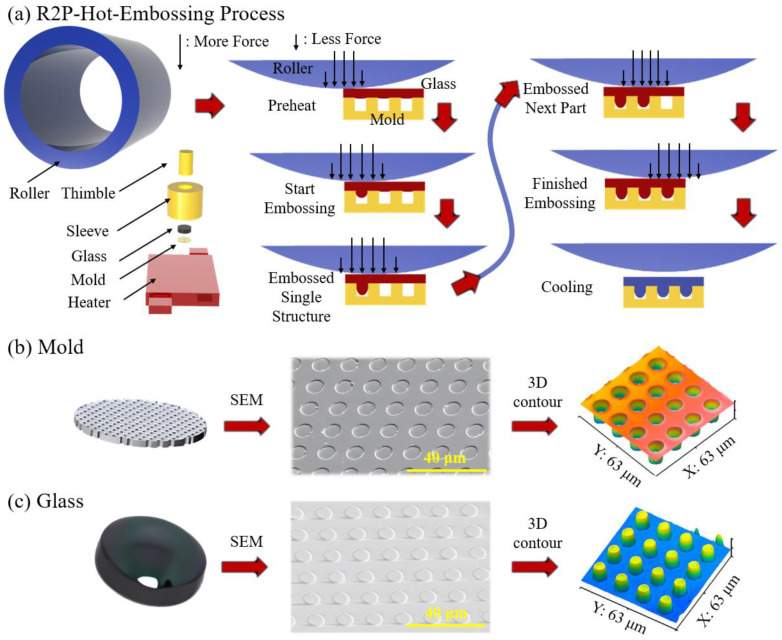
(**a**) Roll-to-plate hot-embossing process; (**b**) mold and its characterization; (**c**) hot-embossed glass and its characterization.

**Figure 2 micromachines-15-01307-f002:**
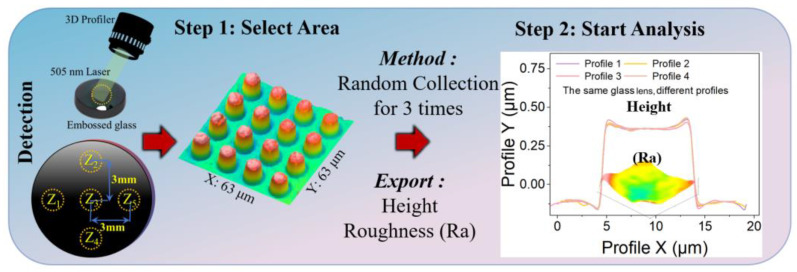
Characterization of structure for roughness and height.

**Figure 3 micromachines-15-01307-f003:**
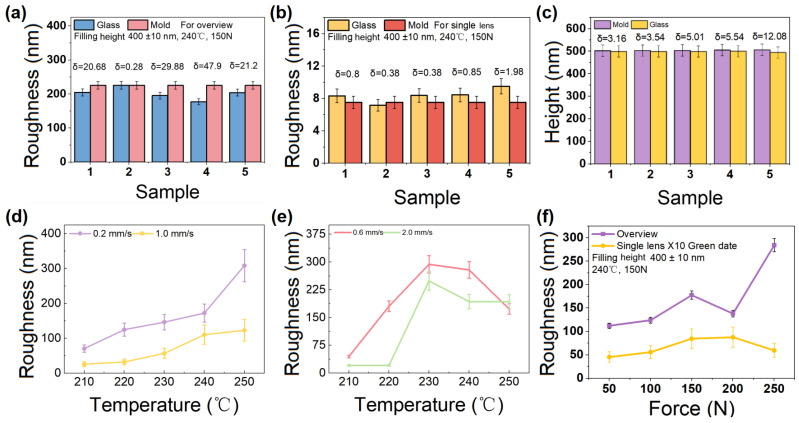
(**a**) Roughness plots of microstructures and mold as a whole; (**b**) plots of average deviation of roughness of individual microlens arrays and individual mold holes; (**c**) plots of deviation of mold depth and microstructure height; (**d**) plots of the effect of embossing speed on overall roughness of microstructure arrays; (**e**) plots of the effect of embossing speed on average deviation of roughness of individual microlenses; (**f**) plots of overall roughness and average deviation of roughness of microstructures under different embossing forces, and plot of the effect of embossing speed on the average deviation of roughness of individual microlenses under different embossing forces.

**Figure 4 micromachines-15-01307-f004:**
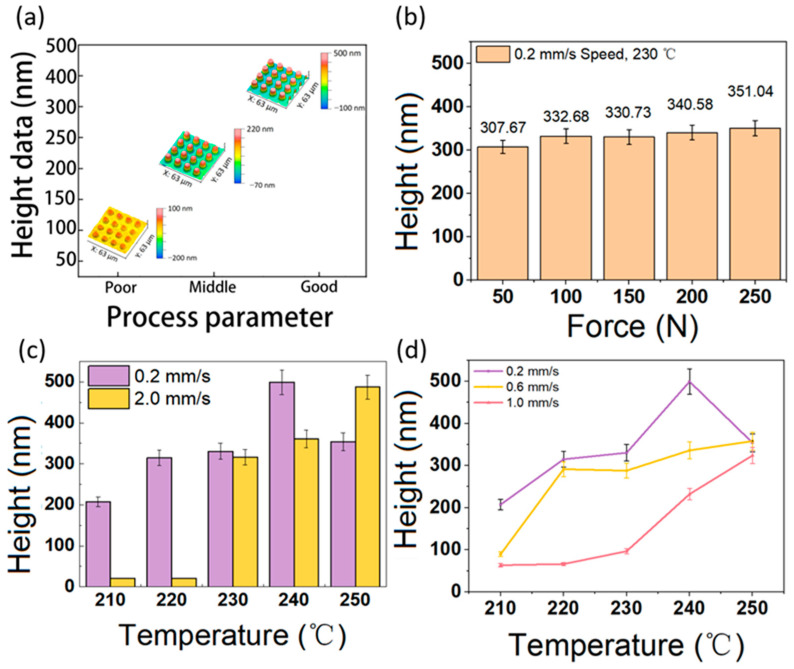
(**a**) Three positional contours of microlenses for three different height ranges; (**b**) variation in microstructure heights for different embossing forces; (**c**) height analysis plots of embossing forces of 0.2 mm/s and 2.0 mm/s; (**d**) variation in microstructure heights for different embossing velocities.

**Figure 5 micromachines-15-01307-f005:**
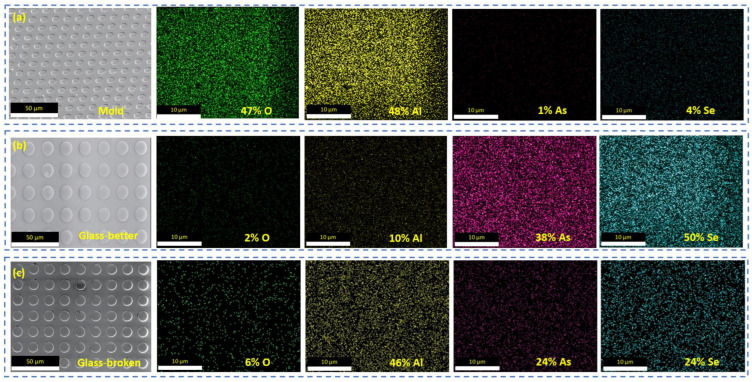
Element distribution analysis of glass surface. (**a**) EDS result for mold; (**b**) EDS result for better glass; (**c**) EDS result for adhesion glass.

**Figure 6 micromachines-15-01307-f006:**
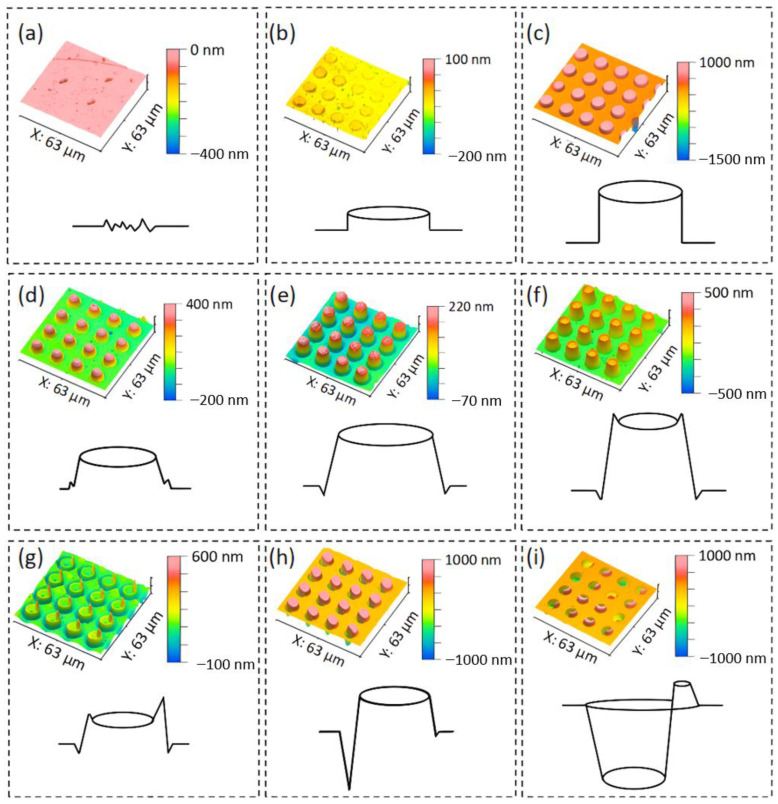
The variation in micron structure height under different processes. (**a**–**c**) Characteristics of microstructure for flat-peak filling mode; (**d**–**f**) characteristics of microstructure for single-peak filling mode; (**g**–**i**) characteristics of microstructure for residual-peak filling mode.

**Table 1 micromachines-15-01307-t001:** Experimental design table for control variables.

Number	Temperature (°C)	Embossing Speed (mm/s)	Embossing Force (N)
1	210	0.2	150
2	210	0.6	150
3	210	1.0	150
4	210	2.0	150
5	220	0.2	150
6	220	0.6	150
7	220	1.0	150
8	220	2.0	150
9	230	0.2	150
10	230	0.6	150
11	230	1.0	150
12	230	2.0	150
13	240	0.2	150
14	240	0.6	150
15	240	1.0	150
16	240	2.0	150
17	250	0.2	150
18	250	0.6	150
19	250	1.0	150
20	250	2.0	150
21	230	0.2	50
22	230	0.2	100
23	230	0.2	150
24	230	0.2	200
25	230	0.2	250

## Data Availability

The original contributions presented in the study are included in the article, further inquiries can be directed to the corresponding author.

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
