# Peer review of "Investigation of an Innovative Roll-to-Plate (R2P) Hot-Embossing Process for Microstructure Arrays of Infrared Glass"

_micromachines, 2024, doi:10.3390/mi15111307_

Round 1
Reviewer 1 Report
Comments and Suggestions for Authors
I expected this study to be an interesting one. However, when I was reading the manuscript, I came across many sentences that are incomplete or repeated twice or have grammatical mistakes or hard to get their meanings, which makes a great burden for the reviewers to understand the significance of this study. Therefore, a thorough re-reading of the manuscript and improvement of the English wring is strongly suggested to the authors. Other specific comments include:
1. The authors mentioned the use of both a femtosecond laser and a picosecond laser within the fabrication processes. Detailed parameters of the laser equipment and processing conditions should be added. Also, what’s specific uses of the femtosecond and picosecond lasers separately?
2. There was only one sentence for describing Fig 1a, “Fig. 1(a) shows roll-to-plate hot embossing process.” (Line 142), which is not sufficient for the reviewer to understand the detailed fabrication processes used in this study. Also, the schematics of Fig 1a are a bit confusing and should be made clearer.
3. The authors mentioned “Individuals denoted microcolumns as Pij, which corresponded to micropores Hij where i = 1, 2, 3, 4 and j = 1, 2, 3, 4.” It seems not necessary to introduce this since it did not appear again in the rest of the manuscript.
4. The Fig 3 and the related descriptions (Lines 189-263) are quite hard to follow and need to be re-organized.
5. Besides, the overall sample roughness from a sampling area of 63 μm × 63 μm seems of little value, since its should be similar with structure height. Therefore, it may be not necessary to show both the overall sample roughness and structure height.
Comments on the Quality of English LanguageA thorough re-reading of the manuscript and improvement of the English wring is strongly suggested to the authors.
Author Response
To Reviewer 1#:
I expected this study to be an interesting one. However, when I was reading the manuscript, I came across many sentences that are incomplete or repeated twice or have grammatical mistakes or hard to get their meanings, which makes a great burden for the reviewers to understand the significance of this study. Therefore, a thorough re-reading of the manuscript and improvement of the English wring is strongly suggested to the authors. Other specific comments include:
Response: Thank you for pointing out the grammar and article errors for me. The author apologizes for these errors. The article has been corrected. The color of the corrected text has been adjusted to red regarding grammatical errors and other issues.
- The authors mentioned the use of both a femtosecond laser and a picosecond laser within the fabrication processes. Detailed parameters of the laser equipment and processing conditions should be added. Also, what’s specific uses of the femtosecond and picosecond lasers separately?
Response: Thank you for pointing out this problem. Specific use of femtosecond/picosecond laser: for the preparation of sapphire molds. The laser equipment is Delong laser processing equipment DP100. The processing process is laser power 10W, laser pulse width 12 picoseconds, frequency 50K Hz, laser wavelength 1064nm, scanning speed 500mm/s.
- There was only one sentence for describing Fig 1a, “Fig. 1(a) shows roll-to-plate hot embossing process.” (Line 142), which is not sufficient for the reviewer to understand the detailed fabrication processes used in this study. Also, the schematics of Fig 1a are a bit confusing and should be made clearer.
Response: Thank you for pointing out that Fig 1a is unclear. Now, Fig.1(a) has a new and more detail about R2P hot embossing process was added. Fig. 1(a) shows roll-to-plate hot embossing process. And more supplementary content was supplied in essay. Displacement of Z axis make roller close to glass. Both roller and glass with the same direction and value of linear speed make emboss can be realized. Then, linear force by roller were conduct in thimble, which can transfer embossing force to glass. Glass partial embossing step by step until embossed were finished.
Fig.1 (a) Roll-to-plate hot embossing process; (b) mold and its characterization; (c) hot embossed glass and its characterization.
- The authors mentioned “Individuals denoted microcolumns as Pij, which corresponded to micropores Hij where i = 1, 2, 3, 4 and j = 1, 2, 3, 4.” It seems not necessary to introduce this since it did not appear again in the rest of the manuscript.
Response: Thanks for your comment of these useless contents, I had deleted these useless contents in essay, including the relevant elements in Fig 2 have also been redrawn.
Fig.2 Characterization of structure for roughness and height.
Fig.2 show the morphology of the embossed glass was measured using a 3D pro-filer (S-nexo, Bruker, Spain). The equipment used was a scanning electron microscope (Titan3 Themis G2, FEI, USA). In order to evaluate the processing accuracy of roll to plate hot embossing process, five zones were sampled for samples at each process parameter. The microstructure of each region was collected in 4×4 arrays for evaluating the microstructural quality. The evaluation consisted of the filling height complex regime and the average roughness (Sa) for two different sampling methods: sample roughness data at the lens, and overall sample roughness data. The overall sample roughness (sampling area: 63μm × 63μm), the roughness of individual lenses (sampling area: 5μm × 5μm), the average deviation of the height of the microlens and its mold average deviation were collected, respectively. In order to obtain a more accurate reproduction rate, we took three measurements at five positions for each bare glass and micrometer structure sample to calculate the data average deviation.
- The Fig 3 and the related descriptions (Lines 189-263) are quite hard to follow and need to be re-organized.
Response: The authors are sorry for the confusing schematic. And, thank you for your pointing these questions in descriptions (Lines 189-263), I had reorganized these manuscript and revise some grammatical questions. About Fig 3 also had been reasonable revised.
Five repetitive experiments were executed before formal experiments. The embossing con-dition is that 240℃ embossing temperature, 150 N embossing force, and 0.2 mm/s embossing speed. After the end of embossing, using the above characterization method. Fig. 3(a) shows the overall roughness in the repetitive experiments, and Fig. 3(b) shows the average deviation of the roughness of individual microlenses and the average deviation of the roughness of individual mold holes. The average deviation of the microstructure of the infrared glass surface from the surface roughness of the mold is 0.878 nm. The surface roughness of the embossed glass is well below the 10 nm deviation from the surface roughness required for general-purpose optical components. Fig. 3(c) demonstrates the error of the average height deviation of the microstructure versus the average deviation of the mold depth for five sets of repeated experiments. Fig.3(c) shows an average deviation of 498.53 nm in structure height with a replication rate of >95%. The results showed that the height deviation of the mold and glass microstructures was a minimum of 3.16 nm, a maximum of 12.08 nm, and an average deviation of 5.862 nm. the R2PHE process demonstrated a satisfactory reproducibility.
Fig. 3(d-e) demonstrates the variation of roughness of the overall and single len micro-structure at different embossing speeds. Together they lead to an increase in the surface rough-ness of the glass. Observing the changes in the average roughness of individual microlenses due to all embossing speeds, it can be assumed that the average roughness of individual microlenses is increasing with the increase of embossing temperature. At increasing embossing temperature, the IR glass and mold surface are under conditions of coupled thermal and force fields. R2P hot embossing process were conducted. In the one hand, glassy selenium arsenide gradually filling molding. In the other hand, molecular inter-penetration becomes more intense between selenium arsenide and aluminum oxide. Materials with different melting points undergo violent inter-penetration. Sometimes, suitable processing temperatures for selenium arsenide do not apply to aluminum oxide. This is severe glass-mold adhesion occur in broken and fault when demoulding (Detail See in Fig.5). Inconsistent coefficients of thermal expansion can also lead to dimensional changes during drastic changes in temperature.
The average deviation of the average roughness of the microlenses with temperature was the smallest for the experimental group is 0.2 mm/s in all the data. The range deviation of 0.2 mm/s was the second. In this data, the less the maximum deviation, the better the uniformity of the microlens is considered to be. Therefore, an embossing speed of 0.2 mm/s is considered to be the embossing speed with the highest potential for the highest quality production.
As the embossing temperature increases, the surface roughness increases in all experimental groups (See in Fig. 3(d-e)). There seems to be no pattern to the fold lines for different embossing speeds. We consider that the micro-vibration of the drum and thimble during the embossing process is the cause. A speed gradient set too small could also be the cause. Separate reviews were conducted for 0.2 mm/s and 1.0 mm/s, or 0.6 mm/s and 2.0 mm/s, respectively. As the speed decreases, the overall surface roughness increases. This is similar to the structural height of the. The surface roughness of the individual microlens decreases as the speed decreases. The experimental group at 250°C does not apply these laws. As you will learn later, this is caused by the phenomenon of adhesion.
Fig. 3(f) shows the variation of the overall roughness of the microstructure and the average roughness of the individual microlenses under different embossing forces (other conditions: 240°C embossing temperature, 0.2 mm/s embossing speed). As can be seen from the figure, both the overall microstructure roughness and the average roughness of individual microlenses show an increasing trend with the increase of embossing force. Both can be considered to be positively correlated with the embossing force. The increase in embossing force increases the contact stress between the upper mold and the glass, and at the same embossing speed (considered to be the equivalent low-frequency vibration frequency), the low-frequency stress amplitude per unit of the microstructure area becomes larger, and thus the microstructure roughness increases both overall and locally. Such a phenomenon is presented more obviously on the whole, while it is not very obvious on the average roughness of a single microlens, and has an obvious effect in Fig. 3(f), which is caused by the fact that the data in the figure are ten times of the original data, which is also to facilitate the representation and comparison of the whole and local situations in one figure.
Fig.3 (a) Roughness plots of microstructures and mold as a whole; (b) Plots of average de-viation of roughness of individual microlens arrays and individual mold holes; (c) Plots of devi-ation of mold depth and microstructures height; (d) Plots of the effect of embossing speed on overall roughness of microstructures arrays; (e) Plots of the effect of embossing speed on average deviation of roughness of individual microlens; (f) Plots of overall roughness and average deviation of roughness of microstructures under different embossing forces and Plot of the effect of embossing speed on the average deviation of roughness of individual microlenses under different embossing forces.
- Besides, the overall sample roughness from a sampling area of 63 μm × 63 μm seems of little value, since its should be similar with structure height. Therefore, it may be not necessary to show both the overall sample roughness and structure height.
Response: Thank you for pointing out this problem. The overall sample roughness from a sampling area of 63 μm × 63 μm were used to confirm that process windows with the condition of adhesion of glass and mold. Due to higher embossing temperature leads to arise with adhesion. These laws had shows in data of the overall sample roughness from a sampling area of 63 μm × 63 μm. As you can see in Fig 3(d-e) and Fig 6(g-i), adhesion make structures of glass are weird. The phenomenon make data with height and roughness abnormal.

Reviewer 2 Report
Comments and Suggestions for Authors
The paper presents some interesting work on developing an innovative roll to plate (R2P) hot embossing process for micro-2 structure arrays of infrared glass. However, the paper manuscript needs to undertake the following revisions:
(1) The paper is better titled as 'Invesitgation of an innovative roll to plate (R2P) hot embossing process for microstructure arrays of infrared glass'.
(2) The Keywords should be rearranged as 'Morphological regulation; Roll to plate hot embossing; Microstructure arrays; Micro manufacturing; Micro-featured surface on Infrared glass'.
(3) In section 2.1, the paper should provide a further clarification and discussion on the design and micro laser processing of micro-structured surfaces on the alumina mold (or roller).
(4) The following very relevant paper in the topic area should be reviewed and included in References section, particularly against the above comments:
- Investigation on an integrated approach to design and micro fly-cutting of micro-structured riblet surfaces, Proceedings of the IMechE, Part C: Journal of Mechanical Engineering Science, 231(18), 2017, 3291-3300.
Author Response
To Reviewer 2#:
The paper presents some interesting work on developing an innovative roll to plate (R2P) hot embossing process for micro-2 structure arrays of infrared glass. However, the paper manuscript needs to undertake the following revisions:
(1) The paper is better titled as 'Investigation of an innovative roll to plate (R2P) hot embossing process for microstructure arrays of infrared glass'.
Response: Thank you for you pointing out this suggestion. I took a hard thinking at the thesis and made a decision. The title of the essay was adjusted to read 'Investigation of an innovative roll to plate (R2P) hot embossing process for microstructure arrays of infrared glass'.
(2) The Keywords should be rearranged as 'Morphological regulation; Roll to plate hot embossing; Microstructure arrays; Micro manufacturing; Micro-featured surface on Infrared glass'.
Response: Thank you for you supply this suggestion. Keywords of the essay were adjusted to read 'Morphological regulation; Roll to plate hot embossing; Microstructure arrays; Micro manufacturing; Micro-featured surface on Infrared glass'.
(3) In section 2.1, the paper should provide a further clarification and discussion on the design and micro laser processing of micro-structured surfaces on the alumina mold (or roller).
Response: Thank you for pointing out this problem. Specific use of femtosecond/picosecond laser: for the preparation of sapphire molds. The laser equipment is Delong laser processing equipment DP100. The processing process is laser power 10W, laser pulse width 12 picoseconds, frequency 50K Hz, laser wavelength 1064nm, scanning speed 500mm/s.
(4) The following very relevant paper in the topic area should be reviewed and included in References section, particularly against the above comments:
- Investigation on an integrated approach to design and micro fly-cutting of micro-structured riblet surface, Proceedings of the IMechE, Part C: Journal of Mechanical Engineering Science, 231(18), 2017, 3291-3300.
Response: Thank you for pointing out this problem. More very relevant papers actually need to be reviewed and included in Reference section. Added references as follows.
Reference:
- Guo Q.; Liu Z.; Yang Z.; Jiang Y.; Sun Y.; Xu J.; et al. Development, challenges and future trends on the fabrication of mi-cro-textured surfaces using milling technology, Journal of Manufacturing Processes, Volume 126,2024, Pages 285-331, ISSN 1526-6125.
- Jiao F.; Sayad Saravi S.; Cheng K.; Investigation on an integrated approach to design and micro fly-cutting of micro-structured riblet surfaces, Proceedings of the IMechE, Part C: Journal of Mechanical Engineering Science, 231(18), 2017, 3291-3300.
- Cao S.; Zhang G.; Huang Z.; Ma Y.; Huo Z.; Du J.; Generation of uniform micro-textured structures by two-dimensional modulation fly-cutting. Journal of Manufacturing Processes, Volume 127, 2024, Pages 238-250, ISSN 1526-6125.

Round 2
Reviewer 1 Report
Comments and Suggestions for Authors
The authors’ efforts to improve the manuscript is appreciated. However, there’s still a big issue with the English writing of this manuscript. Even in the response letter, there are uncompleted and incorrect sentences which greatly weakens the quality of the responses and the manuscript. A thorough improvement of the English wring (with the assistance of native speakers if possible) is again strongly suggested to the authors.
Comments on the Quality of English LanguageA thorough improvement of the English wring (with the assistance of native speakers if possible) is necessary.
Author Response
The revisions have been made according to the reviewer's comments.
Round 3
Reviewer 1 Report
Comments and Suggestions for Authors
In Page 3, it was mentioned: “In this work, femtosecond laser technology was applied to the fabrication of the molds for aluminum oxide…These were made using picosecond laser drilling in a mold.” And in Page 4, it was mentioned: “Femtosecond lasers were used to machine the micro-via arrays on the surface of the sapphire molds, and Delong laser processing equipment DP100 was used. The process involved a laser power of 10 W, a laser pulse width of 12 picoseconds…” If the authors used only a 12 ps laser for their experiments, they should not mention femtosecond laser.
Comments on the Quality of English LanguageThe English wrinting has been much improved.
Author Response

(The authors gave the same response as above.)
